# The Problematic Role of Materialistic Values in the Pursuit of Sustainable Well-Being

**DOI:** 10.3390/ijerph19063673

**Published:** 2022-03-19

**Authors:** Amy Isham, Caroline Verfuerth, Alison Armstrong, Patrick Elf, Birgitta Gatersleben, Tim Jackson

**Affiliations:** 1Centre for the Understanding of Sustainable Prosperity (CUSP), Centre for Environment and Sustainability, University of Surrey, Guildford GU2 7XH, UK; alison@presentminds.org (A.A.); p.elf@mdx.ac.uk (P.E.); b.gatersleben@surrey.ac.uk (B.G.); t.jackson@surrey.ac.uk (T.J.); 2Centre for Climate Change and Social Transformations (CAST), School of Psychology, Cardiff University, Cardiff CF10 3AT, UK; verfuerthc@cardiff.ac.uk; 3Present Minds Ltd., Surrey GU7 3EU, UK; 4Centre for Enterprise and Economic Development Research (CEEDR), Middlesex University Business School, Middlesex University, London NW4 4BT, UK

**Keywords:** materialism, well-being, sustainability, sufficiency, mindfulness, flow

## Abstract

Strong materialistic values help to maintain consumer capitalism, but they can have negative consequences for individual well-being, for social equity and for environmental sustainability. In this paper, we add to the existing literature on the adverse consequences of materialistic values by highlighting their negative association with engagement in attitudes and actions that support the achievement of sustainable well-being. To do this, we explore the links between materialistic values and attitudes towards sufficiency (consuming “just enough”) as well as mindfulness (non-judgmental awareness of the present moment) and flow (total immersion in an activity), which have all been linked to increased well-being and more sustainable behaviours. We present results from three correlational studies that examine the association between materialistic values and sufficiency attitudes (Study 1, *n* = 310), a multi-faceted measure of mindfulness (Study 2, *n* = 468) and the tendency to experience flow (Study 3, *n* = 2000). Results show that materialistic values were negatively associated with sufficiency attitudes, mindfulness, and flow experiences. We conclude with practical considerations and suggest next steps for tackling the problematic aspects of materialism and encouraging the development of sustainable well-being.

## 1. Introduction

The endless pursuit of increasing rates of production and consumption under consumer capitalism has contributed to environmental problems such as the depletion of natural resources, biodiversity loss and climate change enhancement [1,2,3]. Rising material consumption has also failed to consistently improve well-being for people in developed countries [4,5]. Evidence shows that placing high importance on acquiring money and material goods may even be linked to poorer personal well-being [6].

Given the scale of the current environmental crisis, it is important that we place greater emphasis on pursuing our well-being in more sustainable ways. In this research, we explore the attitudes and actions that individuals can take to achieve sustainable well-being, along with the factors that can enhance or hinder our ability to engage in these. In particular, we examine whether strong materialistic values are associated with a reduced likelihood of holding attitudes and engaging in actions that promote sustainable well-being.

### 1.1. Structure of This Work

This work starts by proposing our concept of ‘sustainable well-being’ (Section 1.3), which brings together human well-being alongside care for the environment. We then present three specific ways in which individuals can try to achieve sustainable well-being (Section 1.4). Several authors have proposed that we may be able to lead more fulfilling and more environmentally sustainable lives by consuming less [5,7,8,9]. In terms of the actions that individuals can adopt on their own (e.g., excluding topics relating to community–group initiatives) to achieve sustainable well-being, a handful of suggestions have been proposed in the literature. In this work, we have selected three specific actions that we believe capture the main themes from the existing literature, namely sufficiency, mindfulness, and flow experiences. Each is summarised along with the evidence surrounding their links to both well-being and sustainability in Section 1.4.1, Section 1.4.2 and Section 1.4.3. We also highlight how these three approaches do not have to be considered as mutually exclusive and, instead, can complement each other (Section 1.4.4).

After outlining promising pathways towards sustainable well-being, we go on to introduce the concept of materialistic values (Section 1.5) and theorise around why we expected them to be negatively related to the pathways towards sustainable well-being (Section 1.6). In Section 1.7 we summarise the state of existing knowledge concerning how materialistic values relate to sufficiency, mindfulness, and flow experiences along with how our empirical studies fill the identified research gaps.

Section 2 outlines the methods and findings from three empirical research studies, each testing the cross-sectional relationship between materialistic values and either sufficiency (Section 2.1), mindfulness (Section 2.2), or flow experiences (Section 2.3). In the discussion (Section 3) we highlight the main theoretical research contributions and the practical implications of these for the achievement of sustainable well-being.

### 1.2. Relevance and Novelty of the Research

This research focuses on the interrelationships between environmental health and quality of life for humans. Specifically, it highlights attitudes and actions that can lead to the cooccurrence of environmental health and human well-being (sufficiency, mindfulness, and flow) based on a review of the existing literature. At the same time, it develops our understanding of individual difference factors (materialistic values) that are related to the likelihood of engagement in these potentially beneficial actions through three empirical studies. The research provides novel insights concerning how materialistic values are related to sufficiency, mindfulness, and flow; associations that have previously been unexplored or whose existing evidence base presents methodological limitations. In choosing to examine the differential relationships between subcomponents of materialistic values, rather than just overall scores, the research can more precisely pinpoint specific aspects of materialism that could be particularly problematic for the achievement of environmental and public health.

### 1.3. Sustainable Well-Being

We consider ‘sustainable well-being’ as well-being that has been generated through engagement in actions that have low environmental costs. Well-being in this case covers factors encompassing both subjective (e.g., life satisfaction, positive affect) and eudaimonic (e.g., life meaning and purpose) definitions [10]. By low environmental costs, we mean that the respective activity uses energy and materials at a level that does not risk depleting natural systems beyond the “safe operating space” for the planet [11] and/or promotes care for the environment. O’Brien’s [12,13] conceptualisation of sustainable happiness as “happiness that contributes to individual, community and/or global well-being without exploiting other people, the environment or future generations…” closely aligns with our conceptualisation of sustainable well-being.

### 1.4. Routes towards Sustainable Well-Being

#### 1.4.1. Sufficiency

One way in which the existing literature suggests we may be able to achieve sustainable well-being is simply by acting in more pro-environmental ways [14]. A particular approach people can adopt to engage in more sustainable behaviours is sufficiency. Sufficiency was originally proposed as an economic strategy that focuses on decreased consumption of resources through a reduction in demand for goods [15]. However, as the implementation of sufficiency requires individual level behaviour change [16], psychological approaches have shown to be well-positioned to make valuable contributions to the sufficiency discourse [17,18]. At the individual level, sufficiency is understood as an attitude whereby a person favours more sustainable lifestyle choices [19]. Such voluntary choices are often oriented towards modesty and simplicity [20] and may involves things such as sharing rather than buying new items [21], reducing consumption of resource intensive goods [22] and a preference for less resource intense means of transport such as public transport or biking [16].

Advocates for sufficiency argue that it encompasses a preference for consuming a quantity of goods and services that is enough (or sufficient) for optimal well-being (neither under nor overconsumption) [23]. Princen [24] provides the example of eating to illustrate this point: if we undereat we feel hungry, but if we overeat, we feel bloated. Between these two options there is a point when we are satiated. Eating only enough to be satiated and then stopping would represent the notion of sufficiency. Scholars have argued that sufficiency increases well-being [25,26,27]. Indeed, studies have documented how, when individuals try to consume less, they also tend to report greater well-being [28,29].

#### 1.4.2. Mindfulness

Another promising means of achieving sustainable well-being that is receiving substantial attention in the literature is mindfulness. Mindfulness describes a state of being whereby individuals focus their attention on the present moment [30]. The individual is not distracted by ruminations about the past or hopes and anxieties about the future [31]. When being mindful, people observe their mental states and outside events as they happen on a moment-to-moment basis, and in a non-judgmental manner, not reacting in any automatic or emotionally charged way [32]. Mindfulness is sometimes developed through the practice of meditation, whereby individuals purposefully self-regulate their attention by focusing on internal bodily sensations or sights and sounds in the environment [33]. Nearly any activity can be done mindfully. For example, drinking a cup of coffee ‘mindfully’ could involve focusing attention on how hot the liquid feels on your tongue [34].

Mindfulness has its roots in Buddhist writings where it is presented as a route towards enlightenment and spiritual development [35]. The practice has entered a secular context, sometimes being used as basis for psychotherapy [36] and is the topic of hundreds of self-help books. The well-being benefits of practising mindfulness are well-acknowledged. Mindfulness courses have shown to improve outcomes for patients with disorders such as chronic pain [37] and anxiety [38]. Practicing mindfulness has also been linked to reductions in stress [39] and depression [40], as well as greater life satisfaction [41] and self-esteem [42] in non-clinical populations.

Numerous studies now document a relationship between mindfulness and ecologically sustainable behaviours [43,44,45]. There could be several reasons for this relationship. Firstly, by increasing an individual’s awareness of their habitual thought processes, they can become less susceptible to persuasion from pro-consumerist messages [46]. Secondly, mindfulness has been shown to foster empathy [40] and compassion [47]. Increasing the extent to which individuals can appreciate how their behaviour may be impacting upon other people, and even future generations, can motivate them to behave in more pro-environmental ways [48,49].

#### 1.4.3. Flow Experiences

A third route to achieving sustainable well-being is for individuals to invest all their attention in less energy intensive activities so as to create flow experiences. Flow describes an experience of total immersion that is created when an individual grants all their attention to an activity. During flow, individuals stop perceiving themselves as separate from the actions they are performing, and temporarily lose self-consciousness and track of time. Flow experiences are most likely to occur when there is an optimal matching of levels of challenge versus skill, such that an individual is stretched to perform at their highest level whilst still perceiving themselves as capable of overcoming the respective challenge [50,51]. Whilst theoretically any activity can be moulded to support flow [52], the experience has been shown to be more likely to occur in activities such as schoolwork, craft, exercise, and socialising [53,54,55] and less so in more passive leisure activities such as watching television [56].

As well as being inherently enjoyable, frequent flow experiences are linked to greater life satisfaction [57] and heightened self-esteem [53]. Positive affect can also increase following a single period of flow [55]. Furthermore, neuroscientific evidence has linked the experience of flow to activity in neural reward circuits [58], suggesting that flow can directly influence brain activity associated with positive feelings.

Csikszentmihalyi [59] proposed that the promotion of flow experiences could lead to lower environmental costs. He argues that activities that require low amounts of external energy often also demand large investments of attention, or what he calls ‘psychic energy’. For example, watching television places few mental demands on the viewer but producing and powering the television requires significant amounts of external energy and materials. By contrast, creative writing only needs a pen and paper, but the mental demands on the writer are higher as they must concentrate on generating ideas. Accordingly, argued Csikszentmihalyi, less materially intensive activities tend to require higher investments of attention. As flow experiences require large investments of attention, we should expect them to be more likely to occur in less environmentally costly activities. Indeed, many of the activities that have found to be supportive of flow do not have high environmental costs [53,60]. Research by Isham et al. [54] further supports these ideas by demonstrating a negative relationship between the extent to which US family members reported experiencing flow in an activity and the ‘carbon footprint’ of that activity. The possibility that people are more easily able to find flow in less environmentally costly activities supports the idea that they can achieve a higher well-being with lower environmental impact, since flow is positively correlated with well-being.

#### 1.4.4. Separate Routes or Overlapping Paths?

Above we have outlined three ways that individuals can improve their well-being whilst acting more sustainably. Although they have been outlined separately, the three approaches are not necessarily mutually exclusive (see Figure 1). On the contrary, engagement in one approach may well support successful engagement in another and thus the three actions could operate together within a highly rewarding, sustainable lifestyle. For example, it has been shown that the practice of mindfulness can lead people to discover that their underlying values are not materialistic [61] and, instead, more pro-environmental and/or pro-social and thus they decide to adopt a sufficiency orientation in keeping with this [62].

Likewise, it is known that individuals who are more mindful tend to experience flow more frequently [63], and mindfulness interventions can increase the extent to which athletes experience flow during sport [64]. The practice of controlling attention through mindfulness may develop superior concentration which in turn makes people more likely to experience flow. Although we are not aware of any research exploring the co-occurrence of sufficiency and flow, one might speculate that some of the behaviours that are often undertaken by those individuals with a sufficiency orientation may be more supportive of flow. For example, repairing broken items or coordinating sharing schemes provide higher levels of challenge and require higher levels of skill than does simply repurchasing items.

Accordingly, the three approaches discussed in this article may represent a way of approaching life whereby individuals try to be mindful of or immersed in low environmental impact activities. We examine whether holding strong materialistic values is associated with the extent to which people are likely to adopt this approach to life.

### 1.5. The Problem with Materialistic Values

Individuals holding strong materialistic values place greater importance on acquiring material goods. They consider the acquisition of material goods to be means of improving their own happiness and believe that the number and quality of possessions owned serve as an indicator of their own and other people’s success [65]. In line with this definition, measures of materialistic values include three subcomponents: acquisition centrality (acquiring possessions as key life goal), possession-defined success (use of possessions as criterion for judging own and other’s success) and acquisition as the pursuit of happiness (belief that possessions will boost happiness).

Although materialistic values may help to maintain consumer capitalist economic systems, their negative consequences for individual well-being and sustainability have been well documented. Materialistic values have been linked to negative environmental effects [66]. For example, individuals displaying stronger materialistic values are less likely to donate to environmental charities [67], have higher greenhouse gas emissions [68], and engage less frequently in pro-environmental behaviours such as reusing plastic bags [43]. On top of this, research has shown that highly materialistic individuals report lower levels of personal well-being [6], spanning across components such as lower life satisfaction [69], higher levels of depression and anxiety [70], and a lower sense of purpose in life [71]. These trends appear to operate across income groups and regardless of a nation’s GDP [6].

Recent research has suggested that certain subcomponents of materialistic values may be more problematic for subjective well-being than others. In particular, the acquisition as the pursuit of happiness subcomponent appears to be most detrimental to aspects of well-being such as life satisfaction [72,73,74], general happiness [75], the experience of positive emotions [76], and the satisfaction of psychological needs [77]. Possession-defined success also tends to show negative relationships with well-being, although these are often not as strong as those for the acquisition as the pursuit of happiness subcomponent. The acquisition centrality component, in contrast, has been shown to either not relate to well-being [76] or in some cases to even have a weak positive association with life satisfaction and reductions in loneliness [78,79]. Therefore, it may be that it is people’s beliefs concerning the reasons why possessions are important that drive materialism’s negative association with well-being, rather than simply placing importance on acquiring material goods.

### 1.6. Materialistic Values and Sustainable Well-Being

Whilst existing evidence therefore suggests that materialistic values are linked to both poorer well-being and less sustainable behaviours separately, the evidence surrounding whether materialistic values are also associated with a lesser engagement in attitudes or actions that support sustainable well-being is less developed. We postulate that this might be the case for a couple of reasons. Firstly, materialistic values are extrinsic, or self-enhancement, values [80,81]. Materialism is concerned with the pursuit of one’s own success and happiness over that of others. Self-enhancement values conflict with intrinsic, or self-transcendent, values such as benevolence and universalism which are focused more on the well-being of others and the environment [82]. Strong self-transcendent values have been linked to higher levels of environmental concern, whilst self-enhancement values are associated with less concern for the environment [83]. If materialistic values are associated with lower levels of environmental concern, then we would expect that individuals holding strong materialistic values will be less worried about pursuing their well-being in more sustainable ways.

Secondly, materialistic values appear to be linked to a hedonic orientation to happiness, defining well-being in terms of “pleasure attainment and pain avoidance” [10] (p. 141). Peterson et al. [84] outlined how people could be motivated to pursue their happiness through three routes: hedonism (pleasure), eudaimonism (meaning), and engagement. All three routes are considered as important for achieving the greatest well-being (‘the full life’), but it has been shown that an orientation to meaning and engagement is more strongly related to subjective well-being than an orientation to pleasure [85]. Research [86] has documented that extrinsic values are more strongly linked to a hedonic orientation and people displaying stronger materialistic values have been found to be more likely to engage in hedonic behaviours such as excessive shopping, smoking, and drug use [6]. In contrast, those actions that can support sustainable well-being seem to be more aligned with a eudaimonic or engagement orientation, in that they are often not immediately gratifying, but rather help to build meaning through investments in time and/or effort. For example, research [87] has shown that engagement in ethical consumption behaviours is positively related to eudaimonic well-being but negatively related to a hedonic orientation. Accordingly, a focus on hedonism over eudaimonism or engagement may discourage more materialistic individuals from engaging in actions that support sustainable well-being.

### 1.7. Research Gaps and the Present Research

Following the reviewed literature, there is therefore a theoretical rationale for expecting that strong materialistic values may be negatively related to the tendency to engage in attitudes and actions that we have outlined to be linked to both well-being and sustainable outcomes. However, currently there is little research directly exploring how materialistic values relate to sufficiency, mindfulness, and the experience of flow. In this section, we summarise findings from key literature for each of these three relationships and introduce our three empirical studies that were designed to specifically fill existing research gaps. All three studies employ a cross-sectional survey design. Materialistic values were always assessed using the Material Values Scale (MVS) [65] which is the instrument that examines each of the three proposed subcomponents of materialistic values and has been shown to have good construct validity and psychometric properties [88]. All other constructs were measured using established Likert-response questionnaires.

#### 1.7.1. Materialism and Sufficiency

We are not aware of any research that has directly tested the relationship between the extent to which individuals possess strong materialistic values and their likelihood of adopting a sufficiency attitude. However, it has been suggested that sufficiency is often driven by non-materialistic values [89,90]. Empirical findings have shown that materialism is negatively associated with concepts that are related to sufficiency such as voluntary simplicity [91] and anti-consumption attitudes [92]. Given the lack of existing evidence on the relationship between sufficiency and materialistic values, we designed Study 1 to be the first empirical test of this association. To do this, we employed the well-established MVS [65] in addition to a shortened version of Henn’s [93] sufficiency attitude scale. To the best of our knowledge, this is the only scale to directly test attitudes towards a sufficiency-oriented lifestyle on an individual level. The scale was used by Verfuerth et al. [17] to show that individuals’ sufficiency attitudes were negatively related to their carbon footprint.

#### 1.7.2. Materialism and Mindfulness

Unlike sufficiency, a small number of studies have started examining the association between materialistic values and mindfulness practices. In [94], a negative correlation between materialism and people’s general tendency to be mindful in everyday life (*r* = −0.34) was reported in a sample of Chinese college students. Using the same scales, [95] found a negative correlation (*r* = −0.28) when sampling undergraduate students at a Canadian university and [96] reported a negative correlation (*r* = −0.24) when sampling Italian citizens. However, all former studies employed the MVS [65,97] and the Mindfulness Attention Awareness Scale [26], which has 15 items intended to measure “…the state of being attentive to and aware of what is taking place in the present…” (p. 822). This poses a limitation because mindfulness is more than just a state of being aware. That is, it also encompasses elements of observation, being non-reactive and non-judgemental, amongst others. Therefore, in order not to miss other important facets of mindfulness that may also be linked to materialism, in Study 2 we assess the association between materialism and a more encompassing measure of mindfulness.

#### 1.7.3. Materialism and Flow Experiences

It has been suggested that holding strong materialistic values may prevent an individual from successfully creating flow experiences, but this relationship has only started to be empirically tested in the last couple of years. Reasons for believing that materialism may be negatively related to flow include that a focus on external rewards such as money or praise [80] prevent an individual from engaging in an activity purely because they enjoy it [52] and that a high concern about self-image [98] limits the extent to which individuals let themselves become absorbed by an activity. Two recent studies have examined the relationship between materialism and flow experiences. In [99], a negative correlation (*r* = −.19, *p* < 0.001) between scores of the MVS and a measure of individuals’ general tendency to experience the characteristics of flow in the everyday lives, using an opportunity sample of 451 adults, was reported. The authors of [100] then replicated this finding (*r* = −0.18, *p* < 0.001) using the same measures but this time using a nationally representative sample of 2000 British adults. These studies therefore document that individuals displaying stronger materialistic tendencies seem to be less likely to experience flow. However, one problem with existing correlational studies [99,100] is that their measure of participants’ tendency to experience flow, the Swedish Flow Proneness Questionnaire [101], only includes seven out of the nine proposed characteristics of flow [52]. To increase the validity of the evidence base, it is important to utilise a measure of flow experiences that includes all nine proposed characteristics. Study 3 was therefore designed to test the association between materialistic values and an alternative measure of flow proneness which does include all proposed characteristics.

#### 1.7.4. Differential Relations across Materialistic Value Subcomponents

We outlined how existing findings document that certain subcomponents of materialistic values may be more problematic for well-being than others. In particular, the acquisition as the pursuit of happiness subcomponent appears to be most detrimental to well-being, followed by possession-defined success [72,73,74,75,76,77]. The acquisition centrality component does not appear to be as problematic for well-being [76] and has even, in some cases, been shown to have a weak positive association with certain aspects of well-being [78,79]. However, research rarely explicitly examines how the different subcomponents of materialistic values may be related to factors that can lead to higher well-being. Only [99] tested for differences in the strength of each subcomponent’s associations with flow proneness, where they found that the happiness subcomponent displayed a stronger negative relationship with flow proneness than did acquisition centrality. By utilizing the MVS [65,97] in each of our three empirical studies we were therefore able to produce new insights surrounding the differential associations across materialism subcomponents for each of the three sustainable well-being concepts.

## 2. Empirical Studies

### 2.1. Study 1: Materialism and Sufficiency

The aim of Study 1 is to test the hypothesis that materialistic values will be negatively related to sufficiency attitudes. It also examines the differential relations between the three materialistic value subcomponents and sufficiency attitudes.

#### 2.1.1. Materials and Methods

An online questionnaire was distributed through email lists, Facebook groups, and to undergraduate psychology students at a University in Germany. The topics of the Facebook groups varied. Some focussed on vegan food or sustainable living, whereas most groups were platforms for trading and selling goods and services. People were also invited to forward the study link to their friends and colleagues. Psychology undergraduates participated as part of their requirement to collect research participation hours. All other participants could enter into a prize draw to win EUR 20. The questionnaire was framed as a study on sustainability and personal attitudes.

In total, 310 people completed the questionnaire. Of these, 216 were female and 94 were male. Participants’ mean age was 26.99 years (SD = 7.95, range 17–65). Regarding the education of the sample, 89% of the participants held an A-level or university degree. Of all participants, 62.7% were students, conducting an apprenticeship, or pupils. A further 29.7% were working and 2.8% were unemployed. Most participants had an income between EUR 500 and EUR 1000 per month (37.4%).

To measure attitudes towards sufficiency, a short version of the sufficiency attitude scale [92] was used. The 6-item scale covers a variety of attitudes and opinions about waste of resources, frugal lifestyle, and oversupply of consumer goods. Example items include “Through my lifestyle I want to use as little resources as possible (e.g., water, energy, wood)” and “I find it desirable to possess few things only.” Participants were asked to state how much they agreed with the statements on a scale from 1 (strongly disagree) to 6 (strongly agree). In the present study, the scale demonstrated good reliability, α= 0.84. Materialistic values were measured with the German version of the 15-item Material Values Scale [97,102]. The instrument assesses the three proposed subcomponents of materialistic values: centrality (“The things I own aren’t all that important to me”, α = 0.68), success (“The things I own say a lot about how well I’m doing in life”, α = 0.76), and happiness (“I’d be happier if I could afford to buy more things”, α = 0.83). Participants rated their agreement with each item on a scale from 1 (strongly disagree) to 5 (strongly agree). Here, the full scale showed good reliability, α = 0.85.

#### 2.1.2. Results

The overall association between materialistic values and sufficiency attitudes was first tested using a linear regression analysis (see Table 1) which controlled for age, gender, education, and income. This revealed a large, negative relationship between materialistic values and sufficiency attitudes (*f*^2^ = 0.57) [103]. The change in *R*^2^ value with the addition of the materialism predictor was significant (*F*(1, 275) = 155.95, *p* < 0.001).

To test the differential relationships between the three MVS subcomponents and sufficiency attitudes we conducted tests of equality of correlation coefficients [104]. These highlighted that the correlation coefficient between sufficiency attitudes and the happiness subcomponent of materialism was significantly less negative than the coefficient between sufficiency attitudes and the centrality (z = 2.77, *p* < 0.01) and success (z = 3.50, *p* < 0.001) subcomponents of materialism. The centrality and success subcomponent did not significant differ in the extent to which they were correlated with sufficiency attitudes. See Table 2 for the correlations between all variables included in this study.

Accordingly, Study 1 supports the hypothesis that individuals holding strong materialistic values are less likely to adopt a sufficiency attitude. We believe that this is the first empirical study to document this negative association. It appears that the centrality and success subcomponents of materialistic values are more strongly related with a lack of sufficiency attitude than the happiness subcomponent.

### 2.2. Study 2: Materialism and Mindfulness

The aim of Study 2 is to test the hypothesis that materialistic values will be negatively related to a multifaceted measure of mindfulness. It also examines the differential relations between the three materialistic value subcomponents and mindfulness.

#### 2.2.1. Materials and Methods

To ensure both diversity and specificity within the sample several recruitment strategies were applied. For diversity, the questionnaire was sent to 1000 randomly selected households from the local telephone directory in the UK. To increase the number of participants who were likely to be involved in a mindfulness style of meditation, 60 questionnaires were left at two meetings of a Buddhist meditation group in the southeast of England. Finally, the questionnaire was sent in online format to the whole population of the local university. Participants were given freepost return envelopes and offered the chance to enter a prize draw that gave the opportunity to win one of four GBP 25 store vouchers.

A total of 493 responses were received, of which 25 were deleted because they did not answer any of the materialism or mindfulness questionnaire items. The final sample consisted of 468 respondents: 186 from the random households, 30 from the meditation groups and 252 from the University population. The mean age across the whole sample was 47.8 years (SD = 16.7, range 18–92). A total of 187 respondents were male, 259 female and 22 did not state their gender. The level of education was higher than average with nearly half of the respondents indicating that they possess a post-graduate level of education (45.7%). Half (50.1%) of those who stated, were in full-time employment. Mean household income levels were in the range of GBP 40k–50k.

Materialistic values were measured using the 18-item version MVS [65,97]. As in Study 1, this assesses the three proposed subcomponents of materialistic values: centrality (“I enjoy spending money on things that aren’t practical”, α = 0.71), success (“I admire people who own expensive homes, cars, and clothes” α = 0.76), and happiness (“My life would be better if I owned certain things I don’t have” α = 0.79). Participants rated their agreement with each statement on a scale of 1 (strongly disagree) to 5 (strongly agree). In this study, the full scale showed good internal reliability, α = 0.84. Mindfulness was measured using the Five Facet Mindfulness Questionnaire (FFMQ) [105]. This scale is composed of five facets, or subcomponents. These are non-reactivity to inner experience (“In difficult situations, I can pause without immediately reacting”, α = 0.79), observing/noticing/attending to sensations/perception/thoughts/feelings (“I notice how foods and drinks affect my thoughts, bodily sensations, and emotions”, α = 0.77), acting with awareness/concentration/non-distraction (“I rush through activities without being really attentive to them”, α = 0.87), describing/labelling with words (“I can easily put my beliefs, opinions, and expectations into words”, α = 0.86) and non-judging of experience (“I tell myself I shouldn’t be thinking the way I’m thinking”, α = 0.88). Participants rated how generally true each statement was for them on a scale from 1 (never or very rarely true) to 5 (almost always or always true). The full scale showed good internal reliability, α = 0.87.

#### 2.2.2. Results

A linear regression analysis first tested the relationship between overall materialism and overall mindfulness, controlling for the effects of gender, age, education, and household income (see Table 3). This demonstrated that there was a small but significant effect of materialistic values on overall mindfulness (*f*^2^ = 0.06). The change in *R*^2^ value with the addition of the materialism predictor was significant (*F*(1, 310) = 6.10, *p* < 0.05). To examine relationships between the subcomponents of the mindfulness scale and the materialism scale we tested for equality of correlation coefficients [104]. These tests revealed that overall materialism was not significantly negatively correlated with all facets of mindfulness (see Table 4). In particular, the correlation coefficient between overall materialism and non-reacting was significantly less negative than the coefficient between materialism and act with awareness (z = 2.00, *p* < 0.05) and materialism and non-judging (z = 2.31, *p* < 0.05). None of the other mindfulness facets significantly differed from each other in the extent to which they were correlated with overall materialism.

Tests for equality of correlation coefficients were also carried out to determine whether all three subcomponents of materialistic values were equally associated with overall mindfulness. These tests revealed that there were no significant differences between the strength of the correlation between overall mindfulness and the three materialism subcomponents.

Study 2 supports existing findings surrounding the negative relationship between materialistic values and mindfulness [94,95,96]. It has also built on these findings by showing that there are certain aspects of being mindful that seem to be more strongly associated with materialism than others. In particular, it seems that more materialistic individuals struggle not to judge their own experiences and also to be aware/attentive to their experiences and the activities they engage in. One further benefit of this study was that it employed a wider sample than previous research. Existing studies have focused mainly on student samples, but here we have shown that the negative relationship still exists when our sample also includes more frequent meditators and local households. There appeared to be no differences in the extent to which the three subcomponents of materialistic values were related to overall mindfulness.

### 2.3. Study 3: Materialism and Flow Experiences

The aim of Study 3 is to test the hypothesis that materialistic values will be negatively related to a measure of flow experiences that includes all nine proposed characteristics of flow. It also examines the differential relations between the three materialistic value subcomponents and flow experiences.

#### 2.3.1. Materials and Methods

A nationally representative sample of 2000 adults in the United Kingdom was recruited via a market research company. Quotas were implemented concerning age, gender, socioeconomic status, and geographical region using the latest available government census data. The questionnaire was completed online, and the order of the individual questionnaires randomised.

Materialistic values were measured using the 15-item version of the MVS [65,97]. Five items were used to represent each of the three proposed subcomponents of materialistic values: acquisition centrality (α = 0.60), acquisition as the pursuit of happiness (α = 0.72) and possession-defined success (α = 0.78). The overall scale showed good reliability in the present study (α = 0.83). Participants’ tendency to experience flow was assessed using the Short Dispositional Flow Scale 2 (Short DFS2) [106] which is a nine-item scale tapping into all of Csikszentmihalyi’s proposed components of flow. These are challenge–skill balance, action–awareness merging, clear goals, unambiguous feedback, concentration on the task at hand, sense of control, loss of self-consciousness, time transformation, and intrinsically motivating. As the Short DFS2 is normally completed in relation to a specific activity, we had participants complete the scale three times: in relation to their work/study activities (α = 0.87), leisure activities (α = 0.87), and household chores (α = 0.89). They rated, on a scale of 1 (never) to 5 (always), how often they experienced each characteristic, in general, when engaged in each of the three types of activities. Averaging across these three contexts was intended to provide a score reflecting their general tendency to experience flow (Ullén et al.’s [101] Swedish Flow Proneness Questionnaire also asks participants to respond across these three activity categories to get a measure of people’s general tendency to experience flow in their daily lives). This scale showed excellent reliability in the present study (α = 0.94).

#### 2.3.2. Results

The overall relationship between materialistic values and flow proneness was first tested in a linear regression model which controlled for age, gender, education, and socioeconomic status. This test (results outlined in Table 5) demonstrated that there was a small, negative relationship between materialistic values and the tendency to experience flow (*f*^2^ = 0.02). The change in *R*^2^ value with the addition of the materialism predictor was significant (*F*(1, 1991) = 36.02, *p* < 0.01).

To examine differences across the three subcomponents of materialism and flow, we again conducted tests for equality of correlation coefficients. These revealed that the correlation coefficient between flow proneness and the success subcomponent of materialism was significantly less negative than the coefficient between flow proneness and the centrality (z = 4.01, *p* < 0.001) and happiness (z = 7.64, *p* < 0.001) subcomponents of materialism. The correlation coefficient between flow proneness and the centrality subcomponent of materialism was also significantly less negative than the coefficient between flow proneness and the happiness subcomponent of materialism (z = 3.30, *p* < 0.001). See Table 6 for correlations between all variables in this study.

Study 3 has therefore been able to support existing findings surrounding the negative relationship between materialistic values and the tendency to experience flow, this time using an alternative measure of flow proneness. It has demonstrated that the strength of individuals’ materialistic values is negatively correlated with their general tendency to experience flow. The acquisition as the pursuit of happiness subcomponent of materialistic values was most strongly negatively related to flow proneness, followed by acquisition centrality. The possession-defined success subcomponent of materialistic values alone was not significantly related to flow proneness.

### 2.4. Summary of Results across Studies

To synthesise the three empirical studies and their results, Table 7 summarises their rationale, methods, and key findings.

## 3. Overall Discussion

This research has explored three approaches (sufficiency, mindfulness, and the experience of flow) that individuals can take to achieve sustainable well-being and examined how each of these is related to the possession of strong materialistic values. Strong materialistic values have previously been linked to lower levels of personal well-being and to higher environmental costs [6,68]. However, it has not been clear whether such values may also be problematic for the implementation of individual strategies to achieve sustainable well-being. This is important to determine if we are serious about achieving sustainability targets such as the Paris Climate Agreement and Sustainable Development Goals while ensuring that people can live a prosperous, fulfilling life.

### 3.1. Key Findings

This paper has two key empirical findings. The first is that materialistic values appear to be consistently, negatively linked to engagement in the three approaches for achieving sustainable well-being. Study 1 demonstrated that individuals who hold strong materialistic values tend to be less likely to adopt sufficiency attitudes. Study 2 found a small but significant negative association between materialistic values and mindfulness. Study 3 showed that the strength of materialistic values is negatively related to the tendency to experience flow. The second key finding concerns the differential relationships between the materialistic value subcomponents and sufficiency, mindfulness, and flow. Across the three studies, there were varying findings concerning which component of materialistic values had the strongest, negative relationship with the approaches to sustainable well-being. The centrality and success subcomponents had the strongest negative association with sufficiency whilst the happiness subcomponent had the strongest negative association with flow proneness. Furthermore, there were no differences in the strength of the associations between the materialism subcomponents and overall mindfulness. This highlights the importance of taking a more nuanced approach to the study of materialism which considers which particular aspects may be particularly problematic across contexts.

### 3.2. Theoretical Implications

Our findings add to the existing literature on the detrimental consequences of holding strong materialistic values for both personal well-being and ecological sustainability. Whilst the negative association between materialistic values and various components of personal well-being (e.g., life satisfaction, self-esteem, positive affect, flourishing) is well-documented [72,73,74,75,76,77], less work has been done on the specific ways, outside the realm of trying to acquire material goods, in which individuals with strong materialistic tendencies may or may not pursue higher levels of well-being. The orientations to happiness literature [84,86] has made a valuable contribution in showing that extrinsic values, in general, are linked to a hedonic (rather than eudaimonic or engagement) orientation to happiness. In this research, we have further built on these findings by documenting that materialistic values are negatively related to several different actions that are considered to reflect a eudaimonic or engagement orientation.

Existing research has suggested that the acquisition as the pursuit of happiness subcomponent of materialism is most detrimental to individual well-being, whilst the acquisition centrality subcomponent may even have a positive association with well-being [76]. In this research we tested whether each of the subcomponents of materialism may be differentially related to the three proposed routes to sustainable well-being. However, we found inconsistent results across Studies 1–3. Whilst the acquisition as the pursuit of happiness subcomponent did show the strongest negative link to flow proneness, this same subcomponent had the weakest association with sufficiency attitudes. Meanwhile, there were no significant differences in the extent to which the three materialism subcomponents were related to mindfulness. Accordingly, it appears that the specific subcomponents of materialism are not necessarily linked to engagement in actions that promote sustainable well-being in the same way that they are to scores on measures of personal well-being.

Prior research into the associations between materialism and sustainability has largely employed direct measures of environmental attitudes and behaviours such as donating to environmental charities [67] and frequency of pro-environmental behaviours [43]. These behaviours are primarily motivated by pro-environmental values [107] and as such materialistic values are expected to be negatively related to such attitudes and behaviours given the conflict between self-enhancement and self-transcendent values [82]. In this research, we are also concerned with the antecedents of pro-environmental behaviour. However, it may be worth noting that environmental motivations are less central to the three psychological strategies on which we have focused. For example, mindfulness is often pursued out of a desire for self-improvement [108]. Likewise, the experience of flow generates its own psychological rewards [55,58]. Therefore, we add to the understanding of the relationship between materialism and sustainability by showing that materialistic values impede motivations for engaging in more sustainable behaviours, other than those based purely on environmental concerns. The negative implications of materialistic values for sustainability may therefore be more far-reaching than the existing evidence base suggests.

### 3.3. Practical Implications

If materialistic values are associated with a reduced likelihood of engaging in beneficial actions that can support the achievement of sustainable well-being, it raises the question of which steps can be taken to reduce the prevalence of materialism in the population. There are several approaches that can be taken here which have been nicely reviewed by other authors [109]. These include encouraging intrinsic/self-transcendent values which conflict with materialism [82,110], limiting the prominence of consumer advertising in public spaces [111], and reducing feelings of insecurity so that individuals are less inclined to seek security through materialism [112,113].

It is also important to consider the underlying societal drivers that promote the adoption of materialistic lifestyles and keep us trapped in what Jackson [5] calls the ‘iron cage of consumerism’. Capitalist economies are geared towards high levels of labour productivity, meaning that output (and correspondingly, consumer demand) need to continually increase to maintain or increase employment levels. The profit motive leads businesses to strive for the rapid introduction of new and innovative products to remain competitive. Consumption of material goods is a key means through which individuals participate in society and achieve status and, as such, the introduction of novel goods prompts individuals try to ‘keep up with the Joneses’ [114]. Given the environmental and societal problems this continual pursuit of increasing rates of production and consumption present for both well-being and sustainability, it seems of great importance to explore alternative economic models. Research in the fields of degrowth and postgrowth is exploring how economies and societies could be organised without the need for continual productivity and economic growth [115,116] and might provide a potentially fruitful means towards less materialistic lifestyles.

Related to this, it will be important for governments to consider how best they can create a society which gives people opportunities to choose to engage in attitudes and actions such as sufficiency, mindfulness, and flow if they wish to do so. This could involve improvements in access to public transport or greater subsidies for environmentally friendly services and activities. Actions such as these could help to increase the accessibility of sufficiency-related choices and better allow for engagement in activities that could support flow experiences, for example.

### 3.4. Limitations and Future Research

One area for future research is to employ experimental methods to explore whether materialistic values can have direct, causal effects on sufficiency, mindfulness, and flow. Here, we have used cross-sectional methods which can tell us that two concepts are negatively related only. Emerging findings are beginning to suggest that materialistic values may directly limit the experience of flow. For example, [99] showed in two experimental studies that priming materialistic thoughts led participants to report poorer quality flow experiences in a subsequent activity, when compared to a control group. If future studies can show that materialism also directly undermines engagement in mindfulness and sufficiency attitudes, then this will provide further support for the notion that targeting the strength of materialistic values is an effective intervention for promoting sustainable well-being.

Related to this, it would be interesting to assess whether successful engagement in the approaches discussed in this paper is able to have an impact on the strength of personal values. Preliminary findings [61], for instance, have shown that teaching people mindfulness can lead them to reduce the strength of their materialistic values. If the process of pursuing sustainable well-being can in itself orient people away from materialistic or extrinsic values (which have been linked to greater environmental impacts), then this indicates that approaches such as sufficiency, mindfulness and flow could have far-reaching benefits for environmental sustainability and sustainable well-being alike.

One limitation of the present research is the use of three separate samples with different demographic qualities. This makes it difficult to directly compare effect sizes across the three studies and to generalise the findings across different demographic groups. Future studies should seek to replicate the findings outlined in the present research within a single sample and also across studies using different demographic samples.

Future research would also benefit from taking a mediational approach to try to understand the mechanisms through which materialistic values are negatively related to actions that support sustainable well-being. For example, recent research [103] has documented that materialism may be negatively related to dispositional flow because materialistic values are associated with a desire to avoid negative states and low levels of self-regulatory strength. Similarly, we could hypothesise that a desire to avoid negative states may also fuel materialism’s negative relationship with mindfulness. Indeed, Study 2 found that materialistic values were most negatively related with the ‘non-judging’ aspect of mindfulness. Moreover, as hypothesized earlier, different orientations to happiness could be important explanatory variables in the relationship between materialistic values and attempts to achieve sustainable well-being. Locating the specific reasons why materialistic values are negatively linked to the three actions outlined in this research will help to pinpoint more specific areas for interventions, on top of trying to reduce materialistic values in general.

## 4. Conclusions

Strong materialistic values help to maintain consumer capitalism, but they have negative consequences for individual well-being and sustainability. In this research, we sought to extend understanding of the links between materialism, well-being, and sustainability by examining the extent to which strong materialistic values are associated with engagement with three specific psychological strategies for achieving sustainable well-being. We show that it may well be possible to achieve sustainable well-being, given the existing evidence showing how actions such as a sufficiency orientation, mindfulness and flow experiences can enhance well-being in the absence of severe environmental costs. However, across three studies, we also provide new empirical findings showing that strong materialistic values are negatively related to engagement in these actions. This means that materialistic values may have the potential to hinder attempts to transition towards more rewarding, sustainable lifestyles. It is therefore crucial to challenge the dominance of materialistic values and that future research explores the more precise, causal mechanisms through which materialistic values can have their detrimental effects on the achievement of sustainable well-being.

## Figures and Tables

**Figure 1 ijerph-19-03673-f001:**
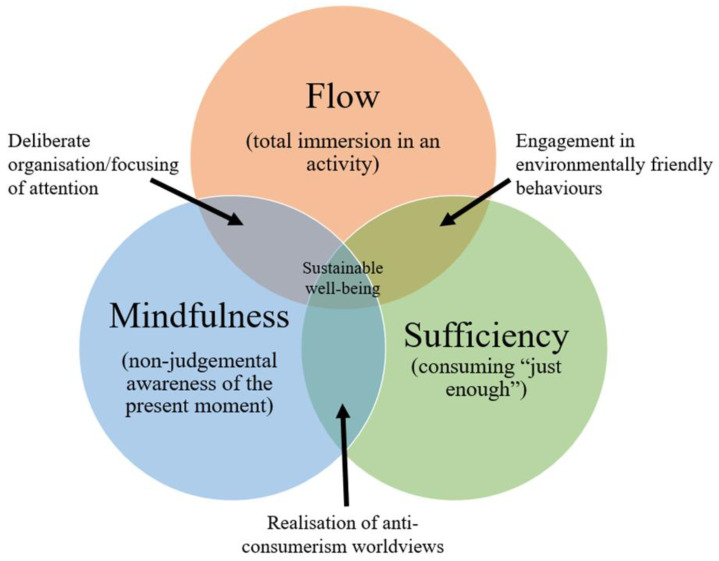
Different individual routes to sustainable well-being and their common features.

**Table 1 ijerph-19-03673-t001:** Linear Regression Analysis examining the influence of the materialistic values on sufficiency attitudes.

	*t*	*p*	β	*F*	*df*	*p*	*R* ^2^
Model 1				6.98	4, 276	<0.001	0.09
Gender (0 = male, 1 = female)	4.82	0.00	0.28				
Age	0.14	0.89	0.01				
Education qualification level	1.76	0.08	0.10				
Income	−0.82	0.41	−0.05				
Model 2				39.83	5, 275	<0.001	0.42
Gender (0 = male, 1 = female)	3.55	0.00	0.17				
Age	−0.69	0.49	−0.04				
Education qualification level	1.22	0.22	0.06				
Income	−1.23	0.22	−0.06				
Materialistic values	−12.47	0.00	−0.59				

**Table 2 ijerph-19-03673-t002:** Correlations between all variables in Study 1.

	Age	Educ	Income	MVS Total	MVS C	MVS S	MVS H
Education	0.08						
Income	0.42 **	0.15 **					
MVS total	−0.06	−0.07	−0.02				
MVS centrality	−0.09	−0.03	0.00	0.83 **			
MVS success	−0.03	−0.04	0.06	0.86 **	0.61 **		
MVS happiness	−0.04	−0.10	−0.06	0.87 **	0.58 **	0.61 **	
Sufficiency attitude	−0.03	0.07	−0.13 *	−0.61 **	−0.57 **	−0.59 **	−0.45 **

* *p* < 0.05 ** *p* < 0.01.

**Table 3 ijerph-19-03673-t003:** Linear Regression Analysis examining the influence of the materialistic values on mindfulness.

	*t*	*p*	β	*F*	*df*	*p*	*R* ^2^
Model 1				3.59	4, 325	<0.01	0.04
Gender (0 = male, 1 = female)	0.55	0.59	0.03				
Age	3.33	0.001	0.20				
Education qualification level	2.30	0.02	0.14				
Household income	0.75	0.46	0.04				
Model 2				4.10	5, 315	<0.01	0.06
Gender (0 = male, 1 = female)	0.50	0.62	0.03				
Age	2.77	0.01	0.17				
Education qualification level	1.97	0.05	0.12				
Household income	0.97	0.34	0.06				
Materialistic values	−2.47	0.01	−0.14				

**Table 4 ijerph-19-03673-t004:** Correlations among all variables including facets of mindfulness and materialistic values.

	Age	Educ	Income	MVS Total	MVS C	MVS S	MVS H	Overall Mindful	Aware	Observe	Describe	Non-Judge
Education	−0.39 **											
Household income	−0.03	0.23 **										
MVS total	−0.14 **	−0.11 *	0.05									
MVS centrality	−0.12 *	−0.06	0.18 **	0.75 **								
MVS success	−0.09	−0.07	0.11 *	0.84 **	0.46 **							
MVS happiness	−0.11 *	−0.12 *	−0.20 **	0.73 **	0.27 **	0.45 **						
Overall mindfulness	0.10	0.09	0.09	−0.17 **	−0.12 *	−0.11 *	−0.17 **					
Act with Awareness	0.14 **	0.09	0.08	−0.15 **	−0.15 **	−0.09	−0.11 *	0.61 **				
Observing	0.08	−0.07	−0.10	−0.07	−0.05	−0.10 *	−0.03	0.52 **	0.05			
Describing	0.03	0.18 **	0.13 *	−0.10 *	−0.02	−0.09	−0.14 **	0.69 **	0.27 **	0.27 **		
Non-judging	0.01	0.12 *	0.12 *	−0.17 **	−0.06	−0.16 **	−0.18 **	0.53 **	0.39 **	−0.08	0.19 **	
Non-reacting	0.05	−0.01	0.00	−0.02	−0.07	0.03	−0.03	0.64 **	0.11 *	0.37 **	0.29 **	0.04

* *p* < 0.05 ** *p* < 0.01.

**Table 5 ijerph-19-03673-t005:** Linear Regression Analysis examining the influence of the materialism on flow proneness.

	*t*	*p*	β	*F*	*df*	*p*	*R* ^2^
Model 1				5.83	4, 1992	<0.001	0.01
Gender (0 = male, 1 = female)	0.21	0.84	0.01				
Age	3.40	0.00	0.08				
Education qualification level	0.37	0.71	0.01				
Socioeconomic status ^1^	−2.11	0.04	−0.05				
Model 2				11.95	5, 1991	<0.001	0.03
Gender (0 = male, 1 = female)	−0.02	0.99	0.00				
Age	1.79	0.07	0.04				
Education qualification level	−0.08	0.94	−0.00				
Socioeconomic status ^1^	−2.12	0.03	−0.05				
Materialistic values	−6.00	0.00	−0.14				

^1^ Socioeconomic status was measured by asking participants to indicate the profession of the chief income earner in their household. Lower scores indicated a higher level of profession.

**Table 6 ijerph-19-03673-t006:** Correlations between all variables in Study 3.

	Age	Educ	Socioecon Status ^1^	MVS Total	MVS C	MVS S	MVS H	Overall Flow	Flow: Work/Study	Flow: Leisure
Education	−0.03									
Socioeconomic status ^1^	−0.26 **	−0.35 **								
MVS total	−0.26 **	−0.06 **	0.08 **							
MVS centrality	−0.04	−0.03	−0.00	0.75 **						
MVS success	−0.21 **	−0.04	−0.01	0.85 **	0.50 **					
MVS happiness	−0.34 **	−0.08 **	0.20 **	0.80 **	0.39 **	0.49 **				
Overall flow proneness	0.09 **	0.02	−0.08 **	−0.15 **	−0.13 **	−0.04	−0.21 **			
Flow: work/study	0.12 **	−0.00	−0.02	−0.19 **	−0.20 **	−0.08 **	−0.21 **	0.86 **		
Flow: leisure	0.08 **	0.04	−0.12 **	−0.13 **	−0.10 **	−0.03	−0.20 **	0.89 **	0.67 **	
Flow: household	0.05 *	−0.01	−0.01	−0.12 **	−0.10 **	−0.04	−0.15 **	0.86 **	0.58 **	0.65 **

* *p* < 0.05, ** *p* < 0.01. ^1^ Socioeconomic status was measured by asking participants to indicate the profession of the chief income earner in their household. Lower scores indicated a higher level of profession.

**Table 7 ijerph-19-03673-t007:** Summary of empirical studies and findings.

Approach to Achieving Sustainable Well-Being	Problems with Existing Evidence	Current Examination	Relationship with Overall Materialism	Differential Relationships with Materialism Subcomponents
**Sufficiency attitudes** (Preference for sustainable lifestyle choices such that the individual consumes just enough to achieve optimal well-being)	No existing evidence	Study 1: 310 German adults complete sufficiency attitude scale [93] and MVS [65,97].	Negative association (β = −0.59, *p* < 0.01) demographics controlled for	centrality (*r* = −0.57) = success (*r* = −0.59) > happiness (*r* = −0.45)
**Mindfulness** (Deliberate focusing of attention on the present moment whereby mental states are observed rather than automatically reacted to)	Few studies, and have only focused on a single-faceted measure of mindfulness	Study 2: 493 adults from UK households, university and meditation groups complete MVS [65,97] and Five Facet Mindfulness Questionnaire (FFMQ) [100].	Negative association (β = −0.14, *p* < 0.05) demographics controlled forGreatest negative effects for non-judgemental and acting with awareness aspects of mindfulness	centrality (*r* = −0.12) = success (*r* = −0.11) = happiness (*r* = −0.17)
**Flow experiences** (Dedication of all attention to an activity leading to feelings of total immersion, oneness with the activity, and lack of self-consciousness)	Few studies, and have only utilised measures of flow proneness that do not include all proposed characteristics of flow	Study 3: 2000 adults from UK nationally representative survey completed MVS [65,97] and Short Dispositional Flow Scale 2 (Short DFS2) [105].	Negative association (β = −0.14, *p* < 0.01) demographics controlled for	success (*r* = −0.04) < centrality (*r* = −0.13) < happiness (*r* = −0.21)

## Data Availability

The datasets generated during and/or analysed during the current study are available in the Open Science Framework repository, https://osf.io/yme78/?view_only=6602888659044954aead390df3615901 accessed date: 22 April 2021.

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
