# Peer review of "The Problematic Role of Materialistic Values in the Pursuit of Sustainable Well-Being"

_ijerph, 2022, doi:10.3390/ijerph19063673_

Round 1

Reviewer 1 Report

The paper deals with a relevant topic. The authors have chosen an appropriate methodology and implemented it well. They have defined their goals clearly and worked systematically in their effort to reach them. The references are appropriate and up-to-date.

I find this conclusion of utmost significance: "the values promoted by consumer cultures may hinder attempts to transition towards more rewarding, sustainable lifestyles." As the authors indicate, more studies will need to be conducted to further solidify and interpret this finding. 

Author Response

Thank you for your positive comments on our paper.

To make the methods of our studies clearer to the reader we have added a new Section 1.7 ‘Research Gaps and The Present Research’. Here we introduce the rationale behind each of our studies as well as their design (see for example lines 277-281). We hope that this introduction will make it easier for readers to follow and anticipate the methods section for each of the individual studies that follow. We also include a summary table in Section 2.4 which aims to clearly presents the brief rationale, method, and key findings for each study.

Reviewer 2 Report

The manuscript investigates the relationships between materialistic values and sustainable well-being.
This is an interesting topic, and all the effort made by the authors in an extensive bibliographic search has much merit.
However, the article lacks scientific consistency to be published.

Introduction: Very extensive, in addition there are references that should have been placed in the discussion.
Material and methods: I am not clear about the type of questionnaires used and their validation.
Results: 
The results are not convincingly presented, and the importance and uniqueness of the findings and implications are not clear enough to justify the conclusions. I do not have a clear impression of the main conclusion of the study.

Author Response

Thank you for your comments on our manuscript. In response to these we have made the following changes:

  1. “Introduction: Very extensive, in addition there are references that should have been placed in the discussion.”

We have now included a number of references that are first introduced in the introduction to the discussion section also, when referring back to the theoretical background in relation to our new results. See for example, lines 576, 601, and 606.

  1. “Material and methods: I am not clear about the type of questionnaires used and their validation.”

Within each of the ‘materials and methods’ sections for the three studies we have given example items from each of the questionnaires, their response scales and their reliability coefficient within the present study.

To give a clearer overview of the methods across the three studies for the reader, we have further extended the section on ‘The Present Research’ (lines 277-281) outlining that the method was similar across studies with the Material Values Scale always being correlated with another established Likert-scale questionnaire for each sustainable wellbeing concept.

We have also now included in section 1.7 the summaries of the existing state of the evidence base for each of the three relationships examined. Here we highlight problems with existing scales and thus demonstrate why we chose to use each of the questionnaires in our current research in order to overcome the highlighted problems with previously employed measures.

  1. “The results are not convincingly presented, and the importance and uniqueness of the findings and implications are not clear enough to justify the conclusions. I do not have a clear impression of the main conclusion of the study.”

We have now added new sub-headings into the discussion to more clearly highlight and distinguish between (a) key findings, (b) theoretical implications, and (c) practical implications. Within the ‘key findings’ section we have now outlined the two key empirical findings from the work more clearly.

We have restructured each of our three results sections to more clearly highlight the research question that each analysis was testing. For example, through statements such as “The overall association between materialistic values and sufficiency attitudes was first tested using a linear regression analysis (see Table 1)” and “To test the differential relationships between the three MVS subcomponents and sufficiency attitudes we conducted tests equality of correlation coefficients [104].” (lines 383-395).

We have added a new section “1.2 Relevance and Novelty of the Research” (lines 67 – 81) near the start of the introduction which briefly highlights the importance and novelty of the research.

In addition, we hope that the content in the new section 1.7, which more explicitly demonstrates how our research fills the gaps within the existing evidence base, will help to differentiate our findings from previous work.

A summary of the methods and results of each study is now presented in Table 7.

We have also made some alterations to the wording throughout to ensure not to make conclusions that go beyond the scope of the data. 

Reviewer 3 Report

The research presented by the authors is very interesting and it is according to my opinion pretty much adjusted to the journal´s scope. Congratulations!

Nevertheless, there are some improvements that can/should be mad to turn the reading more attractive to a more general audience.

I would suggest the following:

  • Create a subsection “1.1 Structure of This Work” to map the different sections of the whole work. Here it should be described the major parts of the work.

  • Create a subsection “1.2. Relevance and Novelty of the Research” where clearly links the importance of this research and the scope of the journal. Touch some of the main points of the journal: International Journal of Environmental Research and Public Health. A table of a figure would be helpful here.

  • In the introduction, once the concepts of Sustainable Well-being, Sufficiency, Mindfulness, Flow experiences are just definitions to guide the reader across the upcoming research, I would suggest summarizing the key points of each one of these concepts and present them in a Tabular form. For example, you could create a sub-chapter 1.3 Key concepts.

  • There is no reference to Figure 1 in the document. For example, you could add this information as follows: “ Accordingly, the three approaches (Figure 1) discussed …”

  • Please numerate all the subchapters across the whole work. By doing so the reader will not get lost in the different subjects presented in this work. For example, 1.4 The Problem with Materialistic Values, 1.5 Materialistic Values and Sustainable Well-being, 6 The Present Research.

  • Also, the conducted studies could be packed into one section (2. Studies conducted). Hit would be also beneficial to pre-present the studies in a tabular form.

  • In the conclusions section I suggest to clearly highlight the managerial and academic impacts of the research. For example, 5.1 Managerial Implications, and 5.2 Proposed Model and Researched Literature (academic implications).

Author Response

Thanks very much for your constructive comments on our manuscript! Below we outline how we have implemented each suggestion within the revised manuscript.

  • Create a subsection “1.1 Structure of This Work” to map the different sections of the whole work. Here it should be described the major parts of the work.

We have now added the suggested section to give the reader an initial primer on what to expect from the paper, given that a lot of content is presented. Within this section we include the new heading numbers for each section to guide the reader and allow the reader to follow-up on specific sections if needed.

  • Create a subsection “1.2. Relevance and Novelty of the Research” where clearly links the importance of this research and the scope of the journal. Touch some of the main points of the journal: International Journal of Environmental Research and Public Health. A table of a figure would be helpful here.

Thanks again for the useful recommendation. We have now added this subsection as suggested and included a paragraph highlighting why our research is important and how it provides novel insights that build upon existing research gaps.

  • In the introduction, once the concepts of Sustainable Well-being, Sufficiency, Mindfulness, Flow experiences are just definitions to guide the reader across the upcoming research, I would suggest summarizing the key points of each one of these concepts and present them in a Tabular form. For example, you could create a sub-chapter 1.3 Key concepts.

Another very constructive comment. Upon reflection, we have decided to add a summary table to the end of the results section which includes short definitions of each of the concepts alongside the key results from the three empirical studies.

  • There is no reference to Figure 1 in the document. For example, you could add this information as follows: “ Accordingly, the three approaches (Figure 1) discussed …”

We have now added reference to Figure 1 within the text in Section 1.4.4: “Although they have been outlined separately, the three approaches are not necessarily mutually exclusive (see Figure 1).”

  • Please numerate all the subchapters across the whole work. By doing so the reader will not get lost in the different subjects presented in this work. For example, 1.4 The Problem with Materialistic Values, 1.5 Materialistic Values and Sustainable Well-being, 6 The Present Research.

Thanks for this suggestion. We have now numbered each section within the manuscript to make the structure of the work more logical to follow.

  • Also, the conducted studies could be packed into one section (2. Studies conducted). Hit would be also beneficial to pre-present the studies in a tabular form.

We now include all of the research studies within the new section “2. Empirical Studies”. The introductory content to each study has been moved into a new Section 1.7. We hope that outlining the purpose and wider design of each study prior to going into detail on the exact methods and results in Section 2 will help to give the reader a better idea of what to expect within the respective empirical studies section.

  • In the conclusions section I suggest to clearly highlight the managerial and academic impacts of the research. For example, 5.1 Managerial Implications, and 5.2 Proposed Model and Researched Literature (academic implications).

We have now added new sub-headings into the discussion to more clearly highlight and distinguish between (a) key findings, (b) theoretical implications, and (c) practical implications.

Reviewer 4 Report

This paper is based on a conceptual framework that postulates materialistic values predict sustainable well-being via attitudes toward sufficiency, mindfulness, and flow. Although the framework is reasonable and the results could potentially contribute to important knowledge regarding the development of sustainable well being, each study when considered alone is too simple for a paper and all 3 studies when considered together suffer from serious methodological flaws.

Specifically, the 3 key constructs were examined in the 3 different studies separately, making it impossible to determine the role of one construct in the presence of the other two in predicting sustainable well being. Examining the 3 constructs concurrently in the same study would have been the appropriate approach to test the proposed framework.  In addition, different samples were combined and used in the 3 studies rather haphazardly (study 1: university students in Germany in Study 1; study 2: randomly selected households, potential meditation practitioners, and members of an university in the UK; study 3: a nationally representative sample of 2000 adults in the UK), making it impossible to generalize the findings to any one particular population.

The authors are encouraged to design their future studies on testing their proposed framework taking into account these comments.

Author Response

Thank you for your comments on our manuscript and we appreciate the limitations that you have highlighted. We wish to emphasise that the purpose of our research was not to show that the three constructs could predict sustainable well-being as many of these relationships have already been documented in existing literature. Rather, we were interested in testing how materialistic values related to each of the three constructs. You are absolutely correct that it would have been optimal to be able to test all three constructs within the same study such that effect sizes could have been compared more directly. However, given the infancy of research in this area, we feel that our research can still be used as a springboard to prompt further work and we aim to design future studies using the proposed framework while overcoming some of the limitations of the work at hand. Indeed, we have now noted in the limitations section that the use of different samples impacts the generalisability of the results and should be addressed in future studies (lines 663-667).

One of our key intentions when writing this paper was to further explore the importance of materialistic values for sustainable well-being. Even across three different samples, our findings do still indicate this and support the conclusions presented.

We have also made some alterations to the wording throughout to ensure that do not make conclusions that go beyond the scope of the data. For example, removal of sentences such as “Without a wider transformation of underlying values and beliefs promoted within consumer cultures, it seems likely that individual achievement of sustainable well-being may be significantly restricted.” We also made sure that it is clear that we refer to associations and relationships rather than effects or impacts.

Reviewer 5 Report

Comments

People with strong materialistic values ​​place greater importance on the possession and use of material goods and on success, but this option is rational. This is due to national income equity. This is not linear in every society. Equality is the worst solution for fighting poverty and for fighting sustainability. Each individual decides on what contributes to his or her happiness depending on the circumstances of each moment. If we want a more just collective society, then where is the role and intervention of the State? And the ecological footprint of those who travel by plane? Most people who use air as a means of travel do not have strong materialistic propensities.

The responsibility towards future generations in confronting the environmental impacts resulting from activities in the various sectors of economic activity is, in fact, not a new but more pertinent problem. The Sustainable Development Goals are a global call to action to end poverty, protect the environment and climate, and ensure that people everywhere can enjoy peace and prosperity. These are the objectives for which the United Nations intends to contribute to collectively achieving the 2030 Agenda. The 2030 Agenda is a document that enshrines a union of purposes of the signatory countries of the United Nations, which emerged from a meeting of the Member States in the year 2015 in New York. This agenda is a set of strategies and measures to be taken and exercised to promote sustainable development by 2030. UN Secretary-General António Guterres says that “the 2030 Agenda is our Global Declaration of Interdependence”. However, in the 1990s, UN Member States assumed a responsibility to meet the needs of the present generation without compromising the development capacity of future generations.

On the ground or in practice, the parameters and requirements required by legislation and other standards must be subscribed by each organization, company or other. The control and observance of the stages of production of goods and services, namely the monitoring of people's needs, is done, as a rule, by the efficiency of the control of the resources that all economic agents are obliged, at least from a civic point of view, to exercise. As adages we can present, e.g., the reduction of the use of intermediate goods of strategic environmental value, the gradual elimination of animal tests on raw materials for cosmetic products, the limitation of environmental impacts of packaging and the prompt repair of any incidents. All this together further promotes the continuous improvement of processes throughout the production chain of goods and services because it is supposed to incorporate clean technologies. It is about looking at the environmental issue as a transversal theme to all organizational structures to be included in strategic planning. It encourages new businesses or new business models, taking into account the principles and opportunities offered by sustainability. Only at the beginning of the millennium was the development of the eight (plus seventeen) Sustainable Development Goals indicated by the UN. Social transformations demand their waiting time, so the utopia of some will have to wait a long time.

Sugestions

When the authors speak of Sufficiency, does it mean returning to a system of income distribution that failed and put various peoples in misery for several decades of the 20th century? Will it mean going back to central planning? Limit the market price system as much as possible? The State must complement the private sector and not impose itself with the rigidity that is subtly substantiated and disguised in this article. In fact, the role of the State is not mentioned in the text, a situation that should be reviewed.

With regard to materialist values ​​and the three concepts used, materialism always loses out. In future investigations, compare, e.g., with the price system, with the revealed preference of each consumer, with the well-being and collective happiness.

Existing research has suggested that acquisition as the happiness-seeking subcomponent of materialism is more detrimental to individual well-being, while the acquisition centrality subcomponent may even have a positive association with well-being. Capitalism is not terrifying. Certain elites demonize him without thinking that there is no better base alternative. It is an economic system that aims at profit and the accumulation of wealth. It is based on the private ownership of the means of production, although in its practical form it must coexist with the State, which is also an entrepreneur and producer of goods and services that satisfy a plural society. The means of production can be, e.g., machines, land, or industrial facilities, having the social function of generating income through the labor factor. Without this, what the authors write in the conclusions that “This means that the values ​​promoted by consumer cultures may hinder attempts to transition towards more rewarding, sustainable lifestyles” is neither adaptable nor realistic. I do not agree and I understand that it is a phrase that is too powerful in the context of the investigation.

Author Response

Thank you for your comments on our paper. Below we respond to each suggestion in turn.

  • When the authors speak of Sufficiency, does it mean returning to a system of income distribution that failed and put various peoples in misery for several decades of the 20th century? Will it mean going back to central planning? Limit the market price system as much as possible? The State must complement the private sector and not impose itself with the rigidity that is subtly substantiated and disguised in this article. In fact, the role of the State is not mentioned in the text, a situation that should be reviewed.

This is not the case. We wish to emphasize that the three approaches outlined in the paper are all actions that we suggest can voluntarily be adopted by individuals, and should not necessarily be enforced by the government or other societal actors. This is why it is important to start to consider factors that make individuals more or less likely to choose to engage in these strategies. We have altered the text in some cases to make it clear that all three options are approaches that should be voluntarily adopted.  See for example on line 102 “Such voluntary choices are often oriented…”.

We agree with your suggestion to include some consideration of the role of government. In the ‘practical implications’ section (3.3) we now end by outlining how it will be important for governments to consider implementing changes that make it easier for citizens to engage in those actions that can potentially support sustainable well-being should they choose to do so.

  • With regard to materialist values ​​and the three concepts used, materialism always loses out. In future investigations, compare, e.g., with the price system, with the revealed preference of each consumer, with the well-being and collective happiness.

Thank you for these fruitful suggestions for future work. We are already starting to conduct further work looking at the nuanced relationship between materialistic values and well-being. The suggestion to look at how materialistic values relate to revealed preferences is fascinating and, we believe, currently underexplored in the literature.

  • Existing research has suggested that acquisition as the happiness-seeking subcomponent of materialism is more detrimental to individual well-being, while the acquisition centrality subcomponent may even have a positive association with well-being. Capitalism is not terrifying. Certain elites demonize him without thinking that there is no better base alternative. It is an economic system that aims at profit and the accumulation of wealth. It is based on the private ownership of the means of production, although in its practical form it must coexist with the State, which is also an entrepreneur and producer of goods and services that satisfy a plural society. The means of production can be, e.g., machines, land, or industrial facilities, having the social function of generating income through the labor factor. Without this, what the authors write in the conclusions that “This means that the values ​​promoted by consumer cultures may hinder attempts to transition towards more rewarding, sustainable lifestyles” is neither adaptable nor realistic. I do not agree and I understand that it is a phrase that is too powerful in the context of the investigation.

We want to make it clear that we are not equating materialistic values with capitalism. We only intend to imply that the societal environments within cultures that have consumer capitalism as their dominant economic framework are more likely to foster and support the presence of strong materialistic values in individuals.

In line with this, we have gone through the paper and made alterations to our wording to make it clear that we are only suggesting that materialistic values are more likely to be rife in capitalist societies, rather than always being directly caused by capitalism. We have also made sure not to make conclusions that go beyond our data. For example, we have altered the potentially problematic sentence that you highlighted to “This means that materialistic values may have the potential to hinder attempts to transition towards more rewarding, sustainable lifestyles.” We also removed sentences such as “Without a wider transformation of underlying values and beliefs promoted within consumer cultures, it seems likely that individual achievement of sustainable well-being may be significantly restricted.” Which may have previously implied that it is consumer culture that directly impedes progress towards sustainable well-being.

Round 2

Reviewer 2 Report

I consider that the authors have made a great effort to make the requested modifications, so that the article was understandable to the reader. Especially in the part of methodology and results, arguing reasonably. The authors have made the necessary changes so that the discussion and conclusions are also clear enough.

Reviewer 3 Report

Great improvements in the manuscript! Good job!

Reviewer 4 Report

No further comments.